# Buprenorphine Increases HIV-1 Infection In Vitro but Does Not Reactivate HIV-1 from Latency

**DOI:** 10.3390/v13081472

**Published:** 2021-07-27

**Authors:** Germán Gustavo Gornalusse, Lucia N. Vojtech, Claire N. Levy, Sean M. Hughes, Yeseul Kim, Rogelio Valdez, Urvashi Pandey, Christina Ochsenbauer, Rena Astronomo, Julie McElrath, Florian Hladik

**Affiliations:** 1Vaccine and Infectious Disease Division, Fred Hutchinson Cancer Research Center, Seattle, WA 98109, USA; germag@uw.edu (G.G.G.); luciav@uw.edu (L.N.V.); clairel@uw.edu (C.N.L.); smhughes@uw.edu (S.M.H.); yskim4451@gmail.com (Y.K.); rxv126@case.edu (R.V.); Upandey@fredhutch.org (U.P.); rastrono@fredhutch.org (R.A.); kd@uw.edu (J.M.); 2Departments of Obstetrics and Gynecology, University of Washington, Seattle, WA 98195, USA; 3School of Medicine, Division of Hematology/Oncology, University of Alabama at Birmingham, Birmingham, AL 35233, USA; christinaochsenbauer@uabmc.edu; 4Department of Medicine, University of Washington, Seattle, WA 98195, USA; 5Department of Pathobiology, Global Health and Laboratory Medicine, University of Washington, Seattle, WA 98195, USA

**Keywords:** MAT, buprenorphine, morphine, methadone, opioids receptors, HIV-1 latency, HIV-1 reactivation

## Abstract

Background: medication-assisted treatment (MAT) with buprenorphine is now widely prescribed to treat addiction to heroin and other illicit opioids. There is some evidence that illicit opioids enhance HIV-1 replication and accelerate AIDS pathogenesis, but the effect of buprenorphine is unknown. Methods: we obtained peripheral blood mononuclear cells (PBMCs) from healthy volunteers and cultured them in the presence of morphine, buprenorphine, or methadone. We infected the cells with a replication-competent CCR5-tropic HIV-1 reporter virus encoding a secreted nanoluciferase gene, and measured infection by luciferase activity in the supernatants over time. We also surveyed opioid receptor expression in PBMC, genital epithelial cells and other leukocytes by qPCR and western blotting. Reactivation from latency was assessed in J-Lat 11.1 and U1 cell lines. Results: we did not detect expression of classical opioid receptors in leukocytes, but did find nociception/orphanin FQ receptor (NOP) expression in blood and vaginal lymphocytes as well as genital epithelial cells. In PBMCs, we found that at physiological doses, morphine, and methadone had a variable or no effect on HIV infection, but buprenorphine treatment significantly increased HIV-1 infectivity (median: 8.797-fold increase with 20 nM buprenorphine, eight experiments, range: 3.570–691.9, *p* = 0.0078). Using latently infected cell lines, we did not detect reactivation of latent HIV following treatment with any of the opioid drugs. Conclusions: our results suggest that buprenorphine, in contrast to morphine or methadone, increases the in vitro susceptibility of leukocytes to HIV-1 infection but has no effect on in vitro HIV reactivation. These findings contribute to our understanding how opioids, including those used for MAT, affect HIV infection and reactivation, and can help to inform the choice of MAT for people living with HIV or who are at risk of HIV infection.

## 1. Introduction

Globally, the HIV/AIDS and drug addiction epidemics have a large overlap [1]. Substantial epidemiological data suggest that drug addiction results in higher HIV transmission rates [2,3] as well as faster progression to AIDS [4]. This can be explained by a combination of sociological/behavioral and immunological factors [5]. HIV acquisition is higher in people who inject drugs (PWID) due to needle sharing, the trade of sex for money, failure to use condoms, and multiple high-risk sexual partners. Other sexually transmitted infections (STIs) including gonorrhea, herpes, and chlamydia are also more frequent in PWID [6], further fueling the link between drug use and HIV susceptibility.

Several studies have also documented direct effects of illicit opioids, especially heroin or its direct metabolite morphine, on HIV infection. In vitro treatment with these opioids was shown to increase expression of HIV co-receptors CCR5 [7,8,9] and CXCR4 [7], and to favor HIV replication [8,10,11,12] and reactivation [13,14,15,16]; however, a few studies reported the opposite [17,18]. In rhesus monkeys, opioid dependence increased simian immunodeficiency virus (SIV) replication [12] and viral set point [19]; however, in other studies, opioid treatment slowed SIV progression [20,21]. 

Immunologically, the current consensus is that opioid use results in immunosuppression [22,23,24,25,26,27], which could further enhance HIV disease progression. However, the mechanisms of this remain obscure and not all opioids share the same immunosuppressive characteristics [24]. Fentanyl, loperamide, and beta-endorphin induced interleukin 4 (IL-4) in human T lymphocytes, favoring a T helper anti-inflammatory response [28]. In rhesus monkeys, chronic administration of morphine increased five-fold the number of circulating T regs (FoxP3 + CD25+ cells) and augmented Th17 functional activity [29]. The implantation of morphine pellets in mice led to a marked decrease in B cell proliferation after in vitro stimulation as well as a reduced IL-2 and IL-4 response by T cells [30]. Intrathecal administration of morphine in female patients decreased natural killer (NK) activity [31]; this opioid has also been shown to impair nitric oxide production in macrophages [32] and neutrophil recruitment in the lungs, which favored bacterial burden [33]. In contrast, another study suggested that neither hydromorphone nor codeine possess immunomodulatory activity [34].

To reduce the mental, societal, and biological problems resulting from heroin use and prescription opiate addiction, two long-acting opioid drugs, methadone and buprenorphine, are prescribed for medication-assisted treatment (MAT) [35]. Both medications normalized immune function compared to heroin use [26,36,37]. However, other studies indicate that methadone has detrimental effects, for example dampening antibody responses [38], production of cytokines [27], and reactive oxygen species [39], cell migration [40] and phagocytic activity [38]. In contrast, most studies on buprenorphine suggested it has negligible effect on the immune system [41,42,43,44,45], although a few animal studies showed it impairs NK cell activity, lymphoproliferation and cytokine production [46,47], or affects the level of certain hormones [48]. We know even less about the direct impact of these two medications on the HIV life cycle. One study showed that methadone enhanced HIV infection of fetal microglia and monocyte-derived macrophages [49]. No data are available for buprenorphine and HIV infection. 

Thus, both MAT drugs require further study to help determine whether they are equally beneficial for people living with HIV, or whether one of them could be preferable based on its immunobiological and virological profile. In this study, we focused on the direct effects of morphine, buprenorphine, and methadone on de novo HIV infection and on HIV reactivation from latency. We find that these processes are not affected equally by these three drugs and will discuss whether our results could have clinical implications. 

## 2. Methods

### 2.1. Generation of Replication Competent Nanoluciferase (NanoLuc^®^)-Expressing HIV-1 Reporter Virus

We generated the HIV-1 infectious molecular clone (IMC) vNL-sNLuc.6ATRi-B-Bal.Ecto, a secreted nanoluciferase (sNLuc) reporter virus expressing the Env ectodomain of HIV-1_BaL_ within the NL4-3-derived proviral backbone, based on our previously described HIV-1 proviral constructs encoding either the sNLuc.T2A or LucR.6ATRi reporter cassettes [50,51]. The T2A “ribosomal skip peptide” was replaced with the modified encephalomyocarditis virus (EMCV) 6ATR internal ribosome entry site (IRES) element (6ATRi), which enables physiological Nef expression and function [51,52,53,54]. We replaced the *LucR* reporter with secreted NanoLuc^®^ (Promega, Madison, WI, USA) [50], inserting *sNLuc* ORF upstream of 6ATRi. Upon replication, the sNLuc reporter is secreted into the culture supernatant, facilitating kinetic monitoring of infection [50]. The reporter IMC is replication competent and encodes all the viral open reading frames, allowing for multiple rounds of viral replication. 

#### 2.1.1. Description of Plasmid

The ectodomain of Env BaL (GenBank accession number: AY426110.1) derives from the HIV-1 isolate BaL. In the previously described reporter IMC, pNL-LucR.6ATRi-B.BaL.ecto [51], the *Renilla luciferase* gene (*LucR*) was replaced by InFusion^®^ (Takara Bio USA, Mountain View, CA, USA) cloning methods with the soluble nanoluciferase-expressing *sNLuc* gene. Fusion of the *NLuc* gene to an N-terminal secretion signal generates a secreted, 19.1 kDa, form of the NanoLuc^®^ luciferase, secNLuc (Promega, under limited use label license, Madison, WI, USA). In the current IMC, the *sNLuc* IRES cassette was inserted between the NL4-3 *env* and *nef* genes. The *sNLuc* ORF is located downstream of the stop codon (taa) of *env* and a Kozak sequence (ccacc); it is followed by a 26 nt “spacer”, the IRES element and the *nef* gene. The IRES we used is derived from encephalomyocarditis virus (EMCV) (GenBank: EMCV IRES, NC_001479), contains the “wild type” (A)_6_ (“6A”) bifurcation loop, and encompasses a truncated EMCV IRES fragment (“TR”, nucleotides 399 to 833). The proviral plasmid was generated and provided by Christina Ochsenbauer (University of Alabama at Birmingham, Department of Medicine, Birmingham, AL, USA).

#### 2.1.2. Preparation of Viral Stock

Viral stock was generated by Rena Astronomo and collaborators at the Vaccine and Infectious Disease Division, Fred Hutchinson Cancer Research Center (Seattle, WA, USA). The methods and reagents used for the generation of virus stock and calculation of virus infectivity were described previously [50]. In brief, generation of the vNL-sNLuc.6ATRi-B.Bal.Ecto reporter virus by transfection of proviral DNA into 293T/17 cells (ATCC, Manassas, VA, USA) using Lipofectamine 2000 was done according to the manufacturer’s protocol (Thermo Fisher, Grand Island, NY, USA). 293T/17 cells were maintained at 37 °C in a humidified incubator in an atmosphere of 95% air and 5% CO_2_. Viral supernatants were harvested 60 h post-transfection, clarified at 1200× *g* for 10 min, and frozen at −70 °C. Virus stocks were analyzed for nanoluciferase expression using Nano-glo luciferase (Promega, Madison, WI, USA) and were tittered on sub-confluent TZM-bl cells, ARP-8129 (obtained through the NIH AIDS Reagent Program, Division of AIDS, NIAID, Manassas, VA, USA contributed by John C. Kappes, Xiaoyun Wu, and Tranzyme, Inc., Birmingham, AL, USA). Virus was diluted in DMEM supplemented with 1% FBS and 40 µg/mL DEAE-Dextran and added to cells for 4 h. Growth medium (DMEM, 10% FBS, Pen/Strep, glutamine) was added to the cells and incubated for 48 h. Cell monolayers were fixed (0.8% glutaraldehyde, 2.2% Formaldehyde in DPBS) for 8 min and stained for β-galactosidase expression (4 mM potassium ferricyanide, 4 mM potassium ferrocyanide, 400 µg/mL magnesium chloride, 400 µg/mL X-gal in DPBS) for 2 h. Titer (2.5 × 10^7^ PFU/mL) was calculated by counting “Blue” β-gal expressing cells.

### 2.2. Drugs

Morphine base (catalog 9300-007), (+)-(S)-methadone (catalog 9250-001) and buprenorphine hydrochloride (catalog 9064-001) were supplied from the National Institute of Drug Abuse (NIDA) drug supply program. Drugs were handled and used under a controlled substance use license issued by the Washington State Department of Health (FX60503089) and a controlled substance registration certificate by the U.S. Drug Enforcement Agency (RH0497544). Morphine was solubilized in methanol to make a stock of 8 mM concentration. Methadone was dissolved in water, heated and adjusted to a final 500 µM concentration. Buprenorphine was dissolved in water to a 10 µM stock concentration. 

### 2.3. Cytokines

Recombinant human TNF-α (AF-300-01A) was purchased from PeproTech (Cranbury, NJ, USA).

### 2.4. Cell Lines

#### 2.4.1. TZM-bl Indicator Cells

The following reagent was obtained through the NIH AIDS Reagent Program, Division of AIDS, NIAID, NIH: TZM-bl cells (catalog # 8129, RRID:CVCL_B478) from Dr. John C. Kappes and Dr. Xiaoyun Wu [55]. The TZM-bl cell line was generated from JC.53 cells by introducing separate integrated copies of the luciferase and β-galactosidase genes under control of the HIV-1 promoter. This cell line was grown as a single-cell layer in Dulbecco’s Modified Eagle medium (DMEM) supplemented with 10% heat-inactivated fetal bovine serum (FBS, Nucleus Biologics, San Diego, CA, USA), 100 U/mL penicillin G, 100 µg/mL streptomycin, 2 mM l-glutamine and 25 mM HEPES (all Thermo Fisher Scientific, Waltham, MA USA; “D10” medium); cultures were maintained at 37 °C in a humidified incubator in an atmosphere of 95% air and 5% CO_2_. The TZM-bl cells are covered under the U.S. patent number 6,797,462 issued to Tranzyme Pharma: “Cell-based method and assay for measuring the infectivity and drug sensitivity of immunodeficiency virus”.

#### 2.4.2. J-Lat 11.1 Cells

The J-Lat T lymphocyte clone # 11.1 was obtained from Emilie Besnard (Eric Verdin’s laboratory at Gladstone Institute, San Francisco, CA, USA) [56]. This is a Jurkat-based T cell line containing a full-length integrated HIV-1 genome expressing green fluorescent protein (GFP) upon HIV reactivation. The genome generates incomplete virions due to a frameshift in *env*. This cell line was maintained in RPMI1640 (Thermo Fisher Scientific) supplemented with 10% heat-inactivated FBS, 100 U/mL penicillin G, 100 µg/mL streptomycin, 2 mM l-glutamine and 25 mM HEPES (“R10” medium). Cells were maintained at 37 °C in a humidified incubator, with 5% CO_2_, and split every other day (generally at a 1:4 or 1:5 ratio) to an approximate density of 2.5 × 10^5^ cells/mL.

#### 2.4.3. J-Lat A7 Cells

The J-Lat A7 cell line (“J-Lat Tat-GFP Cells A7”, catalog# ARP-9853-256, RRID:CVCL_1G44) was obtained from the NIH HIV Reagent Program, Division of AIDS, NIAID, NIH (contributed by Erin Verdin [56]). ARP-9853-256 is a Jurkat cell line that bears the integrated retroviral construct LTR-Tat-IRES-GFP. This cell line was maintained in the same medium and conditions as described for J-Lat 11.1 cells. These cells and methods of use are covered by US Patents 7,232,685 and 7,544,467.

#### 2.4.4. U1 Cells

The subclone U1 of the HIV-1 infected U937 monocytic cell line was obtained through the NIH AIDS Reagent Program, Division of AIDS, NIAID (U1, catalog # 165–432, RRID:CVCL_M769), contributed by Thomas Folks (Laboratory of Immunoregulation, National Institute of Allergy and Infectious Diseases, Bethesda, MD, USA) [57]. U937 is a pro-monocytic cell line obtained from a pleural effusion of a two-year-old Caucasian male with diffuse histiocytic lymphoma. U1 is a clonal population of U937 cells chronically infected with HIV-1. Cells were maintained in R10 medium at 37 °C in a humidified incubator, with 95% air/5% CO_2_ and passaged following the NIH AIDS Reagent Program’s guidelines. Generally, cells were split every 3 days, to a density of 1.0 × 10^6^ cells/mL.

#### 2.4.5. Isolation of Epithelial Cells and Leukocytes from the Female Genital Tract, and Monocytes, CD4^+^ T Cells and CD8^+^ T Cells from Peripheral Blood

The protocol for obtaining peripheral blood mononuclear cells (PBMC) and genital tissues from patients was approved by the Institutional Review Boards of the University of Washington and the Fred Hutchinson Cancer Research Center in Seattle, WA, USA, with informed consent signed by each donor. Biopsy samples were obtained from benign hysterectomies performed in adult women at the University of Washington Medical Center (Seattle, WA, USA). Following surgery, tissue blocks were kept in ice-cooled calcium- and magnesium-free phosphate-buffered saline (PBS, Thermo Fisher Scientific) containing 100 U/mL penicillin, 100 μg/mL streptomycin, and 2.5 μg/mL Fungizone (all from Thermo Fisher Scientific) and transported to the laboratory within one hour. We adapted a protocol described elsewhere [58,59]. The deep submucosa was removed with surgical scissors and the remaining mucosa was cut into 5 × 5 mm pieces. Genital epithelial cell lines from vagina, endocervix and ectocervix were generated and expanded in the presence as feeder cells and the Rho kinase inhibitor Y-27632, as described before [60]. To isolate vaginal leukocytes from these tissue pieces, we followed a previously published method [61]. 

CD4^+^ and CD8^+^ T cells were isolated from PBMC by negative selection. PBMCs were labeled with a cocktail of biotin-conjugated monoclonal antibodies and either the non-CD4+ (130-096-533, MACS Miltenyi; to obtain CD4^+^ T cells) or non-CD8+ (130-096-495, MACS Miltenyi; to obtain CD8^+^ T cells) T Cell Microbead Cocktail kit, following the manufacturer’s instructions.

To isolate monocytes, PBMCs were plated in R10 at a density of 4 × 10^6^/mL onto 6-well plates (2.5 mL/well). Plates were incubated for 2 h at 37 °C and cells in suspension and adherent were collected separately, washed and lysed for RNA isolation.

All cultures were maintained at 37 °C in a humidified incubator, with 95% air/5% CO_2_.

#### 2.4.6. Differentiation of Langerhans Cells from Hematopoietic Precursors

Langerhans cells were derived from CD34^+^ cord blood progenitors using an established protocol [62]. In brief, cells were thawed and cultured at 1 × 10^4^/mL/well in 24-well plates in X-VIVO 15 (Lonza, Basel, Switzerland) containing 100 ng/mL GM-CSF (5.6 IU/mg), 20 ng/mL stem cell factor (5 × 10^4^ U/mg), 2.5 ng/mL TNF-α (2 × 10^7^ U/mg), 0.5 ng/mL TGF-β1 (2 × 10^7^ U/mg) and 100 ng/mL Flt3 ligand (Flt3L) (all PeproTech, Cranbury, NJ, USA). Cultures were incubated at 37 °C with 95% air/5% CO_2_ in a humidified environment for 5–8 days without feeding or re-plating. Cell numbers increased by 50–100-fold during this time. Clusters containing proliferating Langerhans cells were purified by gently harvesting cells with a pipette and layering them on top of 6 mL of 7.5% BSA (Sigma-Aldrich, St. Louis, MO, USA) in 15-mL tubes: up to eight wells were loaded per column. After 10 min on ice, single cells in suspension were removed by aspirating the BSA columns until 3.5 mL remained. Clusters were concentrated by centrifugation at 300× *g*, resuspended in growth media and used for RNA isolation.

### 2.5. Western Blotting

Whole cell protein extracts were obtained by treating cellular pellets with NP40 buffer (Sigma-Aldrich) containing protease inhibitor (Roche, Basel, Switzerland) on ice for 30 min followed by 10 min centrifugation at 12,000 rpm 4 °C. Total protein concentrations were determined by Pierce^TM^ BCA assay (Thermo Fisher) following the manufacturer’s instructions. For western blotting, 12.5 µg (U1, J-Lat 11.1, endocervical and ectocervical cells) or 8 µg (PBMC, CD4^+^ and CD8^+^ T cells) of total protein were heated to 95 °C for 5 min in 1× DTT-containing sodium dodecyl sulfate (SDS) sample buffer (NP0007, Thermo Fisher) and electrophoresed at 200 V for 25 min in Bolt^TM^ 4–12% NuPAGE Bis-Tri-polyacrylamide gel (NW04210, Thermo Fisher) followed by transfer to Immobilon polyvinylidene difluoride membranes (LC2002, Thermo Fisher) and blocking for 1 h with 5% blotting-grade blocker (Bio-Rad, Hercules, CA, USA). A pre-stained protein ladder was included alongside the samples for 10–180 kDa MW reference (PageRuler™, Thermo Fisher). Primary monoclonal antibodies used for immunoblotting were: anti-OPRL1 (PA5-70443, RRID:AB_2688687, Thermo Fisher, concentration used: 1 µg/mL) and anti-Calnexin (4F10, MBL M178-3, RRID:AB_10694101, concentration used: 0.2 µg/mL). Both secondary antibodies used were horseradish peroxidase-conjugated. For OPRL1 detection, we used goat-anti-rabbit (SBI System Biosciences (Palo Alto, CA, USA), ExoAB antibody kit, EXOAB-KIT-1, used at a 1:25,000 dilution); for calnexin detection, we used goat-anti-mouse IgG H + L (Catalog: 31430, RRID:AB_228307, Thermo Fisher, used at a 1:2,000 dilution). Blots were developed using the SuperSignal^TM^ West Femto developing kit (Catalog: 34095, Thermo Fisher). Chemiluminescence was acquired on a ChemiDoc MP imager (Bio-Rad).

### 2.6. In Vitro Infection of PBMC with HIV-1 Reporter Virus

The protocol for obtaining PBMC from HIV uninfected healthy donors was approved by the IRB of the Fred Hutchinson Cancer Research Center in Seattle with informed consent signed by each donor. 2–4 × 10^7^ PBMC were thawed, resuspended at 10^6^ cells/mL in R10 and stimulated with phytohemagglutinin P (PHA-P; 4 µg/mL) and interleukin 2 (IL-2; 50 U/mL) in the presence of either morphine (1–100 µM), buprenorphine (2–20 nM), methadone (70 nM-1 µM) or vehicle control for 72 h. Cultures were incubated at 37 °C in T-25 cm^2^ flasks (Corning, Thermo Fisher Scientific) in a humidified incubator containing 5% CO_2_. After incubation with opioids, we determined both the viability (%) and the number of live cells using EasyCyte Guava ViaCount Assay (EMD Millipore, Burlington, MA, USA). Following counting and centrifugation (300× *g* for 5 min), approximately 2 × 10^5^ cells were infected in a total volume of 100 µL of R10 with vNL-sNLuc.6ATRi-B-Bal.Ecto at MOI of 0.1, 0.2 or 0.5; non-infected wells were used as controls. In some experiments, we included four—and in others, eight—independent infections per condition. To monitor background luciferase activity during the first days of culture, we also included wells in which we infected PBMC with the HIV virus at MOI 0.2 in the presence of both 1 µM raltegravir (catalog # 11680) and 1 µM efavirenz (catalog # 4624); both drugs were obtained from the NIH AIDS Reagent Program, Division of AIDS, NIAID. The infections were done in U-bottom 96-well plates (Corning, Thermo Fisher Scientific) for 2.5 h at 37 °C. The cells were washed three times with pre-warmed R10 medium and resuspended in a final volume of 100 µL of R10 supplemented with 50 U/mL IL-2 in the presence or absence of the corresponding opioid. Cells were cultured for the duration of the experiment (up to 10 days post-HIV infection). At the indicated time points (24, 48, 72, and 96 h, days 7 and 10 post-HIV infection), plates were spun (300× *g* for 5 min), approximately 30–40 µL of supernatants were collected, transferred to new 96-well plates and frozen at −80 °C until analysis. Fresh media containing the corresponding opioid were replenished on the same day that aliquots were withdrawn.

### 2.7. Determination of Nanoluciferase Enzymatic Activity

Samples were brought to room temperature and an aliquot of 20 µL of each sample was mixed with 20 µL of 1× Nano-Glo^®^ luciferase assay reagent (Nano-Glo^®^ Luciferase Assay System, Promega) in a white flat-bottom polystyrene 96-well plate (Corning, Sigma-Aldrich). The mixtures were incubated for 10 min in the dark and luminescence was read on an MLX 96 Well Plate Luminometer (1 sec/well, read height: 1 mm, Dynex Technologies, Chantilly, VA, USA) and reported in relative light units (RLU).

### 2.8. TZM-bl Indicator Cell Infection

Flat-bottom 96-well plates were seeded at a density of 6.4 × 10^3^ TZM-bl cells per well in D10 medium (100 µL/well) 1 day before infection. On the day of the infection, supernatants were thawed and diluted 1:50 (or at the indicated dilution) in D10 containing DEAE-dextran (20 µg/mL); 100 µL of each dilution was used per well. We set up five independent infections of TZM-bl cells per condition and included a negative control with no virus to account for β-gal basal activity in non-infected TZM-bl cells. Cultures were incubated for 48 h at 37 °C in a humidified incubator with 95% air/5% CO_2_. At 48 h post-infection, plates were spun once (800 g 5 min), washed twice with 200 µL PBS, and 50 µL of Tropix Lysis buffer (Galacto-Star™ β-Galactosidase Reporter Gene Assay System for Mammalian Cells; Thermo Fisher Scientific) was added directly to each well. Samples were stored in parafilm-sealed plates at 4 °C overnight. To measure β-gal enzymatic activity, 5 or 10 µL of each sample were mixed with 100 µL of Galacto-Star reaction buffer (1:50 dilution of substrate in diluent) per well in a white flat-bottom polystyrene 96-well plate (Corning, Sigma-Aldrich). Reactions were mixed by pipetting and were incubated 30 min at room temperature in the dark. Luminescence was measured as described above. As controls, we included: (a) empty wells (background signals from the plate); (b) wells with only Galacto-Star reaction buffer (background signals from both plate and reagents); and (c) supernatants from non-infected TZM-bl cells (background from basal β-gal activity as well as from both plate and reagents).

### 2.9. Quantification of Opioid Receptor Gene mRNA by RT-qPCR Assay

The pre-designed qPCR assays (IDT Technologies, Coralville, IA, USA), as 6-FAM/ZEN/3IABkFQ-conjugates, were as follows: OPRD1 (Ref Seq # NM_000911, exons: 2–3), OPRM1 (Ref Seq# NM_001145285, exons: 1–2), OPRK1 (Ref Seq # NM_000912, exons: 2–3) and OPRL1 (Ref Seq # NM_000913, exons: 5–6). A pre-designed assay to quantify the copy numbers of the peptidylprolyl isomerase A (*PPIA*) or glyceraldehyde 3-phosphate dehydrogenase (*GAPDH*) gene was included to normalize the data. Negative controls (cDNA reactions without adding reverse transcriptase) were included to control for potential non-specific gDNA amplification. A positive control with total human brain RNA (BioChain) was included. Relative quantification by the ΔCt method was applied to calculate the level of expression of different opioid receptors. Relative expression of each receptor was defined as 2^−Δ*C*t^. Δ*C*_t_ is calculated as: Δ*C*_t_ = *C*_t_ (opioid receptor) − *C*_t_ (PPIA or GAPDH). The total number of cycles ran (i.e., 40 or 45) was assigned to compute the limit of detection (LOD) in those samples with undetectable amplification.

### 2.10. Statistical Analysis

Statistical analysis was done using the GraphPad Prism suite (Prism 8 for Windows 64-bit, v8.3.1, RRID:SCR_002798); each statistical test was specified in the corresponding figure legend. Differences were considered statistically different when *p* < 0.05.

## 3. Results

### 3.1. Opioid Receptor Gene Expression in Circulating and Mucosal Cell Types of Relevance for HIV Infection

Opioid receptor expression on a cell suggests its potential for reactivity to opioid exposure. Therefore, before conducting functional experiments of HIV infection and reactivation in the presence of opioids, we wanted to establish whether cell types of direct or indirect relevance for HIV infection transcribe the genes for classical or non-classical opioid receptors. Peripheral blood mononuclear cells (PBMC) did not significantly transcribe any of the three classical opioid receptor genes, mu opioid receptor (*MOP*), delta opioid receptor (*DOP*), or kappa opioid receptor (*KOP*), whereas these were all highly expressed in control brain tissue (Figure 1A). PBMC did express the non-classical nociceptin opioid receptor gene (*NOP*; also known as nociceptin/orphanin FQ receptor) at approximately 2 logs lower levels than brain (mean of two PBMC donors: 0.018 expression relative to the PPIA gene, brain 1.249 relative expression) (Figure 1A). Of note, we ruled out amplification of potentially contaminating genomic *NOP* DNA because the PCR reaction spanned the junction between *NOP* exons 5 and 6. Monocytes and T lymphocytes purified from PBMC in two additional donors expressed *NOP* equally (Figure 1B). Monocytes, but not T lymphocytes, expressed rudimentary levels of *KOP* as well (Figure 1B). 

To consider the potential effects of opioids on HIV-1 reactivation, we assessed opioid receptor expression in two cellular models of HIV latency: J-Lat 11.1 and U1 cell lines, which are derived from CD4^+^ T cells [56] and promonocytic cells [57], respectively. In both models, the level of relative expression of *NOP* was similar (average: 0.032) and was at least 2 logs higher than *MOP*, *DOP*, or *KOP* (Figure 1C).

Because HIV transmission occurs via mucosal barriers and ongoing infection is likely fueled by processes in the mucosa, we also measured opioid receptor transcripts in leukocytes isolated from the vaginal mucosa (Figure 1D). Vaginal T cells and macrophages did not express *MOP*, *DOP*, or *KOP*, but both expressed *NOP*, with macrophages being ~1 log higher than T cells. We also tested in vitro-differentiated Langerhans cells as a surrogate of mucosal Langerhans/dendritic cells and found they transcribed the *NOP* gene even more strongly than vaginal macrophages (Figure 1D), but were also negative for *MOP*, *DOP*, and *KOP*. Because epithelial cells can produce factors, such as TNF-α, which in turn influence HIV infection in leukocytes [60], we also assessed opioid receptor mRNA expression in primary epithelial cells generated from surgically excised vaginal, endocervical and ectocervical tissues. These cells also expressed *NOP* but not the classical opioid receptors (Figure 1D,E). Control qPCR reactions with genomic DNA or with RT-negative RNA showed no amplification, confirming the specificity of the *NOP* PCR assay. 

Finally, we confirmed the level of expression of NOP at the protein level by performing western blotting with a polyclonal anti-NOP antibody directed towards the middle region of this receptor (Figure 1F). We tested protein extracts isolated from J-Lat 11.1, U1, ectocervical and endocervical cells as well as from PBMCs and their corresponding CD4^+^ and CD8^+^ enriched fractions. In all the samples analyzed, we observed a band with an observed molecular weight slightly higher than the predicted weight (41 kDa), in agreement with post-translational modifications in NOP reported by others [63].

In summary, our data show that several cell types relevant for HIV/AIDS pathogenesis express the non-classical opioid receptor *NOP* both at the mRNA and protein levels but not the three classical opioid receptors. 

### 3.2. Effect of Buprenorphine, Methadone, and Morphine on HIV Replication in PBMC

Our finding of *NOP* gene expression in circulating and mucosal T cells and monocyte/macrophages suggested their potential for direct reactivity to opioids. Therefore, we next wanted to test whether opioid drugs have the capacity to modulate HIV-1 infection and replication in vitro. To easily monitor HIV infection in cell cultures, we generated a replication-competent HIV-1 reporter virus based on a previous construct [50,51]. This virus expresses the HIV.Bal_26_ Env ectodomain in an HIV NL 4.3 background and a secreted form of nanoluciferase encoded by *sNLuc*. Unlike the previously used construct in which *nef* was expressed following a T2A ribosome-skipping peptide, in this vector *nef* was expressed from an internal ribosome entry site (IRES) called 6ATRi (Figure 2A). Since the levels of nef achieved are closer to physiological [51,52,53,54], this construct has been shown to better track in vivo HIV-1 infection. Upon transcription of the integrated provirus, nanoluciferase is synthesized and secreted from the infected cells, allowing to measure viral replication in the supernatants by chemiluminescence. In initial HIV infection titrations using PHA/IL-2 activated PBMC, luciferase signal plateaued beyond an MOI of 0.5 (not shown); therefore, we employed MOIs within the 0.1–0.5 MOI range in all experiments. Figure 2B shows a representative kinetic of HIV reporter virus infection at 0.2 MOI over 10 days of culture.

Using this HIV infection assay, we then started comparisons of the effect of morphine, buprenorphine and methadone on HIV-1 susceptibility and replication. A scheme of the experimental setup is depicted in Figure 3A. In these experiments, we activated PBMC with PHA and IL-2 in the presence of opioids and, following three days of culture, infected them with the HIV reporter virus. To analyze the impact of opioids on HIV susceptibility and early infection, we removed the opioid drugs following infection and cultured the cells only in R10 with low dose IL-2. In other experiments presented further below, we replenished cultures with opioids after infection and maintained exposure over the course of in vitro cell culture, to recapitulate the impact of continuing opioid usage on HIV replication and propagation.

Figure 3B shows a representative experiment with PBMC from one donor. We found that 2 nM buprenorphine increased nanoluciferase activity at 24 and 72 h, at all MOIs tested. For example, at 72 h and MOI 0.5 in this donor, nanoluciferase activity was 7.21 times higher in the presence of 2 nM buprenorphine. However, we observed a trend toward a decrease in nanoluciferase signal at the 20 nM concentration of buprenorphine in this donor. Morphine also trended to enhance nanoluciferase activity, but to a lesser extent than buprenorphine, and only at the higher dosage. For example, at 100 μM, morphine caused a 2.4-fold increase of nanoluciferase signal 72 h after infection with MOI 0.2 HIV-1. Notably, the average luminescence signals across all conditions did not increase significantly between 24 and 72 h post-infection, suggesting that HIV was not yet propagated much by cell-to-cell spread during the first 3 days of infection. 

Many individuals with opioid addiction use methadone rather than buprenorphine for medication-assisted treatment. We set up morphine vs. methadone vs. buprenorphine comparisons in several donors, some of whom were repeated independently (summarized in Figure 3C). The data shown is for a MOI of 0.2, but the direction of the effect was comparable across MOIs. The viability was similar across different treatments (Appendix A). Overall, we observed the most consistent effect of nanoluciferase activity enhancement with 2 nM buprenorphine at 24 h (13 experiments, six different donors, median: 1.540, range: 0.7661–4.120; *p* = 0.0017), followed by 100 μM morphine at 24 h (six experiments, four different donors, median: 1.684, range: 1.037–6.003; *p* = 0.0313). 72 h following infection and drug removal, only 20 nM buprenorphine showed nanoluciferase enhancement (11 experiments, six different donors, median: 1.763, range: 0.4971–4.075, *p* = 0.0362). The data revealed substantial inter-donor variability. Collectively, the results presented in Figure 3 identified a weak early HIV enhancing effect of buprenorphine, an even weaker one for morphine, and no effect for methadone. 

The above experiments spanning 1–3 days following HIV exposure likely reflect binding and entry of HIV, with subsequent release into the culture supernatants. To assess the effect of MAT drugs on HIV propagation in longer-term cultures, we replenished methadone or buprenorphine following HIV infection and maintained them at a constant concentration over the full term of monitoring HIV infection in vitro for 10 days (experimental schema on Figure 4A). Treatment with buprenorphine or methadone did not affect cell viability (Appendix A). The data in Figure 4B depict the fold increase over control for different donors (MOI 0.5), some of whom were repeated independently. On day 4 post-infection, only 20 nM buprenorphine increased nanoluciferase activity in supernatants significantly (10 experiments, eight different donors, median: 7.011; range: 1.529–28.96, *p* = 0.002). On days 8 to 10 post-infection, there was a median 8.797-fold increase in luciferase with 20 nM buprenorphine (eight experiments, eight different donors, range: 3.570–691.9, *p* = 0.0078). Methadone had no effect on HIV infection at any dose or time point. The raw results for these comparisons are shown in Appendix A.

We confirmed that nanoluciferase activity tracked the concentration of infectious units of HIV-1. For this, we infected PBMC from two donors, collected supernatants after 10 days post-infection and assayed them for nanoluciferase activity and for infection of TZM-bl indicator cells. Figure 4C demonstrates that both readings were highly correlated (R^2^ = 0.7993, *p* < 0.0001, *n* = 23 points). Figure 4D shows the results obtained in TZM-bl cells with supernatants collected from four PBMC donors 10 days after HIV-1 infection. In all four PBMC donors treated with 20 nM buprenorphine, we observed an enhancement in β-gal readings, whereas 1 µM methadone, tested in two donors, did not influence HIV replication. Therefore, buprenorphine increases HIV-1 replication as measured by two techniques; this effect was weak at early time points and became more pronounced over time.

### 3.3. Effect of Methadone, Buprenorphine, and Morphine on HIV Reactivation from Latency

Whether the MAT drugs methadone and buprenorphine affect HIV reactivation from latency is unknown. To begin to address this question, we used the latently infected T cell line J-Lat 11.1 [56] and the monocytic cell line U1 [57] to test HIV reactivation by opioids. Both cell lines express *NOP* (Figure 1C). There are two additional reasons why we chose to analyze potential reactivation effects on J-Lat 11.1. First, in our previous work [60], we showed it has less spontaneous reactivation than other J-Lat cells (~10% GFP^+^ in media), but it is more easily reactivatable by latency-reversing agents, including TNF-α. Second, J-Lat 11.1 possesses an HIV structure more similar to the HIV provirus found in vivo, compared to some of the other J-Lats, such as A1 and A7, which contain only a mini-HIV cassette [56]. Figure 5A is a summary of three independent experiments in the T cell line and demonstrates no HIV reactivation from J-Lat 11.1 by either buprenorphine, methadone, or morphine across various concentrations that match plasma levels measured in individuals using drugs. We confirmed the lack of HIV reactivating effects of buprenorphine in a separate clone of J-Lat (A7), even when we used a concentration of 2 µM (Appendix A). Figure 5B shows by representative flow cytometry plots that buprenorphine did not increase the percentage of GFP^+^ J-Lat 11.1 cells at 2 and 20 nM, whereas TNF-ɑ strongly induced GFP expression from the HIV promoter. Buprenorphine also did not enhance TNF-ɑ induced HIV reactivation. Likewise, depicted in Figure 5C, none of the opioid treatments stimulated HIV reactivation from the monocytic cell line U1, as measured by a RT-ddPCR assay for HIV-1 LTR-driven poly(A) copies (since U1 cells do not contain a GFP indicator gene). Again, TNF-α reactivated HIV efficiently from U1 cells. Not surprisingly, none of the opioids induced TNF-α mRNA expression in the U1 cells (Figure 5D). In conclusion, buprenorphine, methadone or morphine did not reactivate HIV-1 in the latently infected J-Lat 11.1 T cells or U1 monocytes.

## 4. Discussion

Medication-assisted treatment of opioid addiction is becoming a widely accepted strategy to stabilize the neuronal system and decrease risk behaviors [64]. With MAT and HIV prevention or treatment converging in the management of opioid addiction, it is imperative to know whether and how MAT impacts HIV acquisition, infection and latent reservoir dynamics; and which of the two most widely used MAT alternatives (buprenorphine/naloxone (Suboxone^®^) or methadone) is optimal in this context. Here, we found that buprenorphine but not methadone enhanced HIV-1 replication in PBMC in vitro, but neither drug triggered virus reactivation in two HIV-1 latency models. Buprenorphine’s effect was likely mediated by the non-classical opioid receptor NOP, because NOP was the only opioid receptor we detected in PBMCs. 

We in fact started our study with profiling the expression of opioid receptors in cell types that are relevant for HIV-1/AIDS pathogenesis. In peripheral blood lymphocytes, we detected expression of only the non-classical NOP receptor (Figure 1A,B), which agrees with prior reports [65,66,67,68,69,70,71]. However, two studies also found DOP expression by circulating lymphocytes [72,73]. It remains to be investigated whether these discrepancies may be explained by different states of T cell subset differentiation and/or activation. We also detected NOP, but not MOP or DOP, in monocyte-enriched populations. Monocytes also expressed very low levels of KOP (Figure 1B). KOP expression has been reported before for monocytes/macrophages [74] and microglia [75]. Broad transcriptomics/proteomics datasets [76,77] support NOP expression in various leukocytes, including the latently HIV-1-infected J-Lat T lymphocyte and U1 monocyte cell lines used in our study. Recently, Lambert et al. demonstrated NOP expression also by polymorphonuclear granulocytes [78]. Interestingly, HIV-1 infection itself was shown to alter the splicing pattern of opioid receptors [79,80]. Further studies will be necessary to unravel the regulation of NOP in leukocyte subpopulations and how this intersects with HIV infection. 

In addition to peripheral leukocytes, we also found expression of NOP in T cells and macrophages isolated from the vaginal mucosa, in in vitro derived Langerhans cells, and in epithelial cells cultured from the vagina, the uterine ectocervix and the uterine endocervix (Figure 1D–F). Studies done in rats by Klukovits et al. showed that NOP ligands inhibited myometrium (uterine) contractility in pregnant rats at term [81,82]. However, to the best of our knowledge, there are no published studies concerning the role of the NOP opioid system for immunity and inflammation in the female reproductive tract, and whether its modulation by MAT drugs influences HIV-1 susceptibility. Our finding that NOP is broadly expressed across the reproductive tract of women warrants such investigations.

Opioid treatment of different cell types has been shown to increase the susceptibility to HIV-1 infection and replication (reviewed in [22]). Morphine has been studied as the paradigmatic example of how an opioid enhances in vitro HIV-1 replication [8,9,10,11,12,14]. The most straightforward proposed mechanism is through upregulation of the HIV-1 coreceptor CCR5 [7,8]. Morphine was also shown to downregulate the level of beta chemokines [8], to modulate the expression of proinflammatory markers (e.g., TNF-α, IL-6) [83], to block CD8-mediated anti-HIV activity [84], to inhibit the expression of anti-HIV miRNAs [16] and to transactivate the HIV-1 LTR [85]. In our study, morphine doses representing a physiological range in humans in vivo [86] did not enhance HIV-1 replication in PBMC (Figure 3C). The discrepancy with published work likely stems from the much higher doses of morphine used in earlier studies. Because our doses match more closely those measured in people who use morphine analogs (1–100 µM) [87,88], we postulate that morphine or heroin use per se, i.e., independently of behavioral factors, does not enhance HIV-1 replication in vivo. 

The virologic activities of the MAT opioid drugs buprenorphine and methadone have hardly been studied. Methadone was reported to enhance HIV infection of macrophages and PBMC [49]. No such study exists for buprenorphine. We tested HIV infection and viral uptake in both buprenorphine and methadone treated, pre-activated PBMCs. Cultures were split to address two questions: the impact of opioids/MAT on HIV susceptibility and early infection (drugs removed during culture) and the impact of continuing drug exposure on HIV replication and propagation (drugs maintained during culture). Our results show that buprenorphine augmented HIV replication even during the first time points when HIV propagation from cell to cell was not yet robust. For example, at 24 h we observed almost a two-fold increase in luciferase reads in the cultures treated with 2 nM buprenorphine (Figure 3B,C). Notably, the magnitude of buprenorphine-mediated enhancement of HIV infection increased substantially towards later time points (days 4 and 10) when MAT was maintained during culture (Figure 4B–D). We observed no enhancement of HIV replication by methadone.

The mechanisms whereby buprenorphine enhances HIV-1 replication remain unclear. Two in vivo studies showed that buprenorphine treatment altered systemic cytokine levels, favoring a pro-inflammatory milieu [46,48]. This may create a higher permissiveness for HIV-1 replication. In our in vitro experiments we used PBMC, which include immune cells like monocytes and CD8^+^ T cells that could react to buprenorphine treatment and indirectly enhance HIV-1 infectivity of the CD4^+^ T cells. On the other hand, buprenorphine has been shown to decrease chemokine-induced chemotaxis of monocytes, which led to the hypothesis that it could protect the central nervous system against HIV-associated neurodegenerative disease [89,90]. While these studies suggest a beneficial effect of buprenorphine in chronic HIV infection, they did demonstrate clear and immediate intracellular effects of buprenorphine in immune cells. Specifically, buprenorphine decreased the phosphorylation status of cytoskeletal proteins [90] and the association of cytoplasmic adaptor proteins with the intracellular moieties of CCR2 [89] and CCR5 [91]. It remains to be seen whether and how these activities may enhance HIV replication during the acute phase of infection. 

Based on our opioid receptor expression profile, we presume that the enhancement of HIV-1 replication in buprenorphine-treated PBMC stems from its activity on the NOP receptor. Buprenorphine’s agonistic activity on NOP was previously reported [92] (and reviewed in [93]). Notably, we observed significant inter-individual variation in the extent of responsiveness to buprenorphine (Figure 3C and Figure 4B,D). Analogous to the finding that some individuals do not respond to the analgesic effect of buprenorphine due to a single nucleotide polymorphism in the *MOP* gene [94], it is possible that the inter-donor variation in buprenorphine’s enhancement of HIV replication stems from allelic variation in the *NOP* locus. *NOP* gene polymorphisms have been reported before [95,96], but it still needs to be assessed whether any of these influence the effect of buprenorphine on immune cell function and HIV infection.

There is conflicting data regarding the effects of opioids on HIV-1 reactivation from latency [3]. We found that neither buprenorphine, morphine nor methadone reactivated HIV-1 in the latently infected J-Lat 11.1 or U1 lines at plasma concentrations equivalent to those reported during in vivo use [86,97,98,99]. We also measured the viability of these cells as well as the levels of TNF-α and CCR5 transcripts and did not find any changes following treatment with morphine or buprenorphine. Our results agree with those by Prottengeier et al. [13] who showed that heroin or morphine did not increase HIV p24 expression in latently infected T lymphoblasts at micromolar or submicromolar concentrations; HIV-1 reactivation was only evident when these opioids were added at >1 mM concentration. The authors showed that the opioid-mediated HIV reactivation at these high doses was likely related to cellular necrosis, was prevented by the addition of antioxidants, such as N-acetyl-cysteine, and was not mediated by opioid receptors. 

Our data and Prottengeier’s differ from other studies reporting that morphine [14,15] and methadone [49] could reactivate HIV-1. The discrepancies could be due to several experimental differences between the studies, including the cellular model of HIV-1 latency (chronically infected cell lines vs. cocultures with brain/microglial cells vs. latently infected PBMC from HIV-positive patients), culture conditions (PHA/IL-2 vs. anti-CD3 vs. LPS stimulation), and timing (6 days vs. 3 days). Our in vitro results favor the hypothesis that opioids do not affect HIV latency reversal in vivo. This notion is supported by a clinical study which found no association between buprenorphine treatment and viral load trends in HIV-1-infected individuals on highly active antiretroviral treatment (HAART) [100]. However, more longitudinal studies will be required to definitively rule out an effect of MAT drugs on HIV reservoir dynamics.

In summary, when tested at treatment-equivalent doses, morphine and methadone had negligible effects on HIV infection in PBMCs, while buprenorphine enhanced primary HIV-1 replication. The mechanisms of this effect remain unknown, but likely stem from buprenorphine’s partially agonistic activity via the non-classical NOP receptor expressed by lymphocytes. No opioid drugs tested here enhanced HIV-1 reactivation from latency. These findings suggest that in individuals living with HIV controlled by ART drugs, choice of MAT is unlikely to affect HIV dynamics. However, the use of buprenorphine-containing MAT in those at risk of primary HIV-1 infection may enhance initial viral replication, which underscores the need for effective HIV prevention in former PWID on MAT.

## Figures and Tables

**Figure 1 viruses-13-01472-f001:**
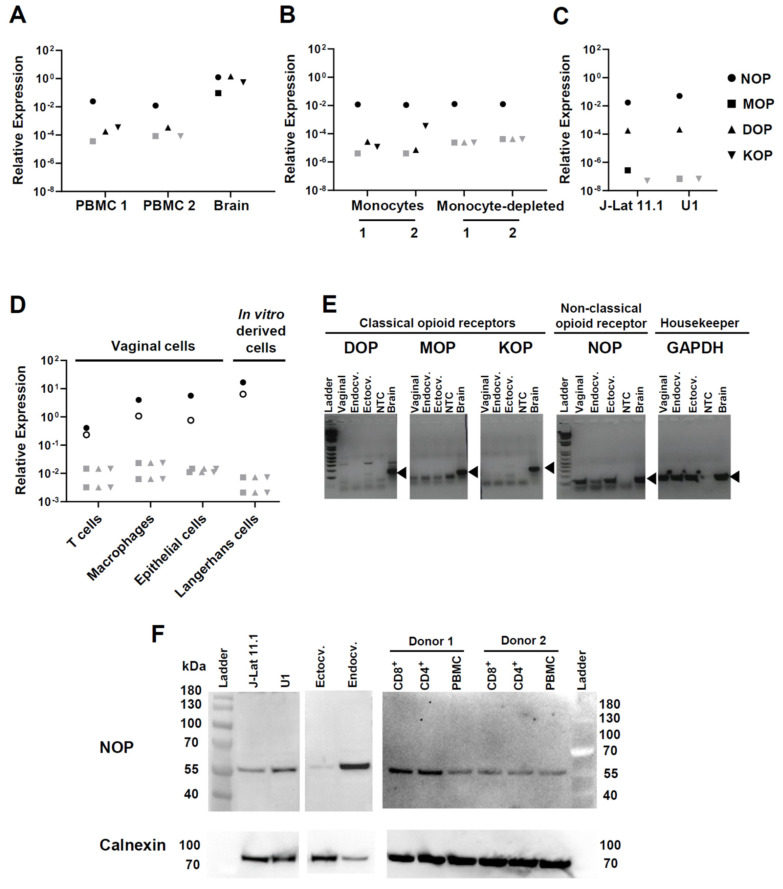
Expression of opioid receptors in different primary cells. RNA isolated from the indicated cell types was assessed for expression of opioid receptors by qPCR. NOP indicates the non-classical nociceptin/orphanin FQ (N/OFQ) peptide (NOP) receptor (OPRL1); MOP indicates the mu receptor (OPRM1); DOP the delta receptor (OPRD1) and KOP the kappa receptor (OPRK1). Relative expression of each receptor was defined as 2^−Δ*C*t^. Δ*C*_t_ is calculated as: Δ*C*_t_ = *C*_t_ (opioid receptor) − *C*_t_ (PPIA or GAPDH). Each symbol represents the average of technical duplicates. (**A**) PBMC were from two healthy donors, and brain RNA was purchased. Opioid receptor expression on the *y*-axis is relative to the housekeeping gene PPIA. Similar results were obtained when GAPDH was used as normalizer (not shown). For a given opioid receptor qPCR analysis, samples with negative amplification (i.e., *C*_t_ > 40) are denoted with gray shaded symbols. (**B**) Gene expression analysis was done in the monocyte-enriched and monocyte-depleted fractions derived from PBMC of two additional healthy donors. (**C**) Gene expression analysis in the two cellular models of HIV-1 latency: J-Lat clone 11.1 and U1 promonocytic cells. Two independent cultures of each cell line were analyzed separately; symbols represent the average of these cultures. (**D**) Vaginal T cells, macrophages, and epithelial cells were sorted from healthy vaginal tissues from two donors, and Langerhans cells were differentiated in vitro from hematopoietic precursors, as explained in Materials and Methods. Data represent gene expression levels per receptor and per donor (each donor is represented with either an open or close circle). Samples with negative amplification (i.e., *C*_t_ > 40) are denoted with gray shaded symbols. (**E**) Gel electrophoresis analysis of qPCR products generated from genital epithelial cell lines. Real-time PCR amplicons were resolved on a 2% agarose gel. Ladder: 100 bp molecular weight marker; Endocv.: endocervical; Ectocv.: ectocervical; NTC: non-template (water) control. Specific PCR amplicon for each gene is indicated with a black triangle. PPIA was used as housekeeping gene in A and D, GAPDH in B, C and E. (**F**) NOP expression was confirmed by western-blotting. Protein extracts were obtained from the two HIV-1 latency models (J-Lat 11.1 and U1 cell lines), from endocervical and ectocervical cell lines, from primary unfractionated PBMC, and from magnetically-purified CD4^+^ and CD8^+^ T cells isolated from two donors. Calnexin is shown as loading control.

**Figure 2 viruses-13-01472-f002:**
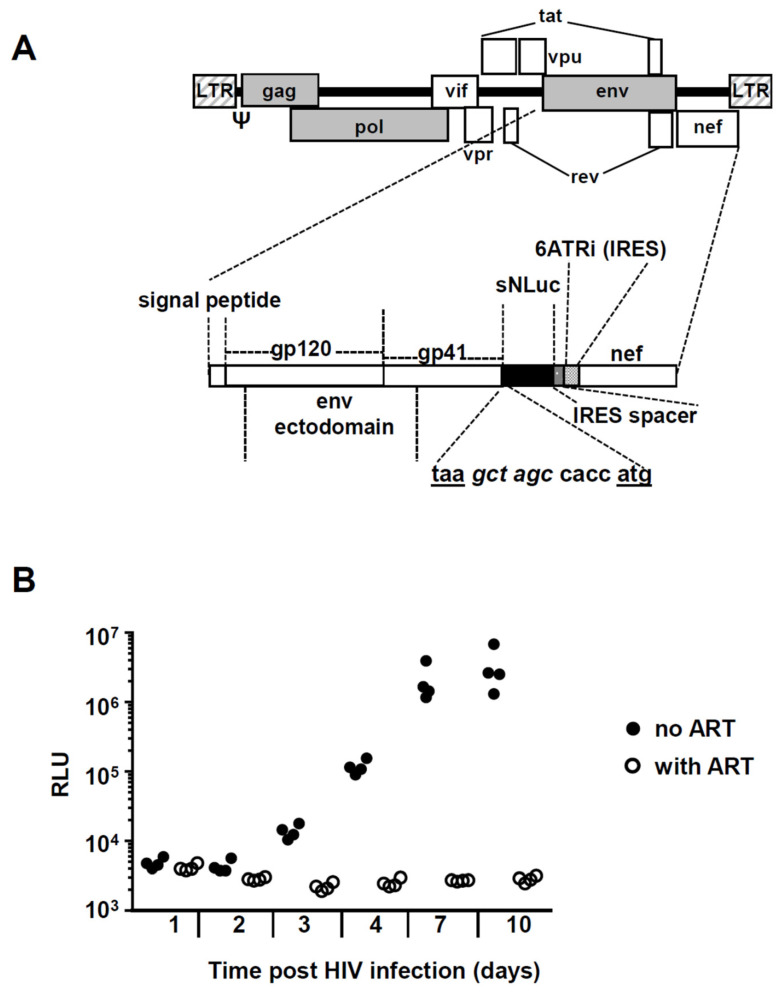
Experimental strategy to quantify in vitro HIV-1 replication in PBMC. (**A**) Simplified schema of the replication-competent R5-tropic HIV-1 reporter virus vNL-sNLuc.6ATRi-B-Bal.ecto. The vector was based on the NL-Bal.Env.ecto-derived molecular clone of HIV-1 expressing Renilla luciferase previously generated by Ochsenbauer and Astronomo et al. [50,51]. The reporter gene (indicated as the black area) is a secreted form of nanoluciferase (encoded by the reporter gene *sNLuc*); nef expression is regulated by an internal ribosome entry site (IRES). The sNLuc.IRES cassette was inserted between the NL.Bal.ecto *env* and *nef* genes; 6ATRi is a truncation (“TR”) fragment derived from the encephalomyocarditis virus (EMCV) IRES and contains the “wild type” (A)_6_ (i.e., “6A”) bifurcation loop. The nucleotide sequence shows the junction between the stop codon of env (taa) and the start codon (atg) of sNLuc, and contains a Nhe I restriction site (gct agc) and the translation initiation Kozak sequence (ccacc). The construct contains a 26 nt “IRES spacer” between the sNLuc gene and the IRES element, as previously described [51]. (**B**) Example of an in vitro HIV-1 infection experiment. PBMC were infected at MOI 0.2 in the presence (empty circles) or absence (full circles) of 1 µM raltegravir and 1 µM efavirenz. Aliquots from supernatants were taken at the indicated time points (*x*-axis) and nanoluciferase activity was determined using the Promega Nano-Glo© kit (*y*-axis, RLU: relative light units).

**Figure 3 viruses-13-01472-f003:**
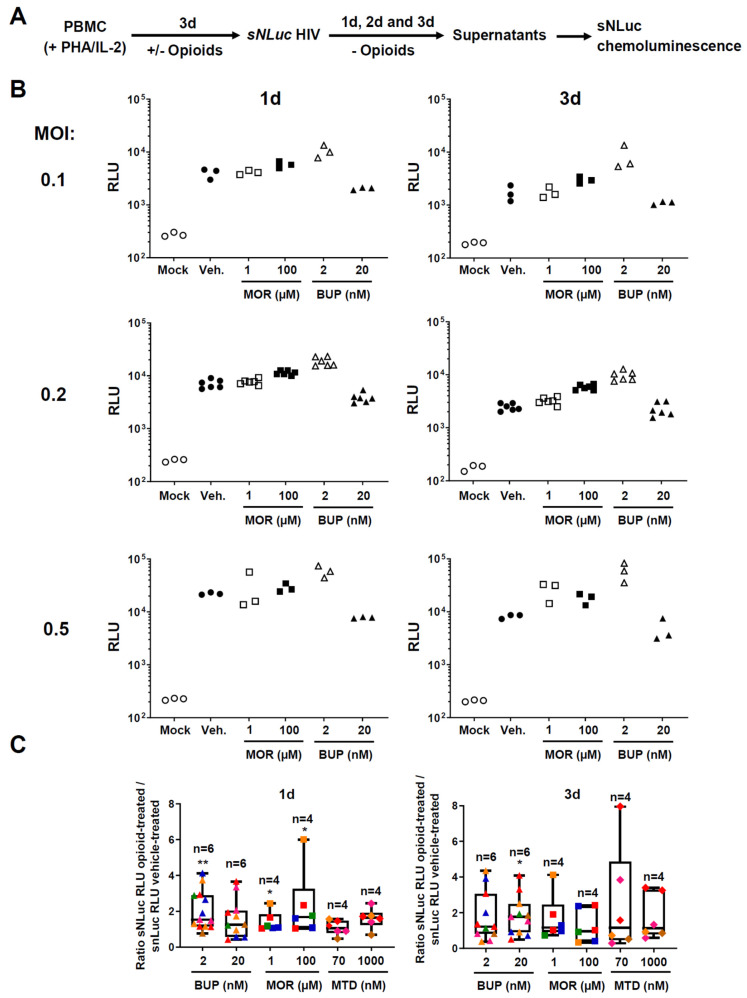
Comparison of the effects of different doses of morphine, buprenorphine and methadone during early HIV-1 infection of PBMC. (**A**) Experimental approach to assess in vitro HIV-1 infection and replication. PBMC were incubated with opioids for 72 h and after infection and washes, drugs were withdrawn. Aliquots were taken at the indicated timepoints, and HIV-1 replication was measured by quantifying nanoluciferase enzymatic activity in the supernatants. Chemiluminescence signals were quantified using a luminometer. (**B**) Representative nanoluciferase experiment done with PBMC from one donor. We cultured PBMC for up to 72 h with PHA/IL-2 and morphine (1 or 100 µM), buprenorphine (2 or 20 nM), or vehicle control. In these experiments, the vehicle control consisted of R10 medium supplemented with PHA, IL-2 and 1.25% *v*/*v* of methanol. The viability, assessed by trypan-blue staining or Guava Muse^®^ analyzer, was similar across different conditions (Appendix A). We infected them with sNLuc HIV-1 and opioids were washed off. Luminescence signals were determined with supernatants taken at 1 and 3 days after HIV-1 infection. *Y*-axis denotes relative light units (RLU); number next to *Y*-axis indicates the Multiplicity of Infection (MOI) of the HIV-1 reporter virus used to infect the PBMC. Each treatment is represented as a different symbol (specified in the *X*-axis); open and close symbols denote the lowest and highest dose for each drug, respectively. We did a total of three technical replicates for MOI of 0.1 and 0.5, and 6 for MOI 0.2. (**C**) Summary plots showing the effect of opioids on nanoluciferase activity measured in supernatants taken at 1 and 3 days after HIV-1 infection. The MOI here was 0.2; we did six technical replicates per condition. Each donor is represented by a different color; the number of different donors per treatment is indicated with an “*n*” above the plots. *X*-axis indicates different treatments and concentrations. *Y*-axis shows the ratio of luciferase activity measured in opioid-treated cultures over luciferase activity measured in vehicle-treated cultures. Plots indicate median (vertical lines in the middle), interquartile range (boxes), and min to max range (whiskers). *p*-values were two-tailed and calculated using the Wilcoxon Signed Rank Test, comparing the median obtained in each treatment with a theoretical median value of 1. *: *p* < 0.05; **: *p* < 0.01.

**Figure 4 viruses-13-01472-f004:**
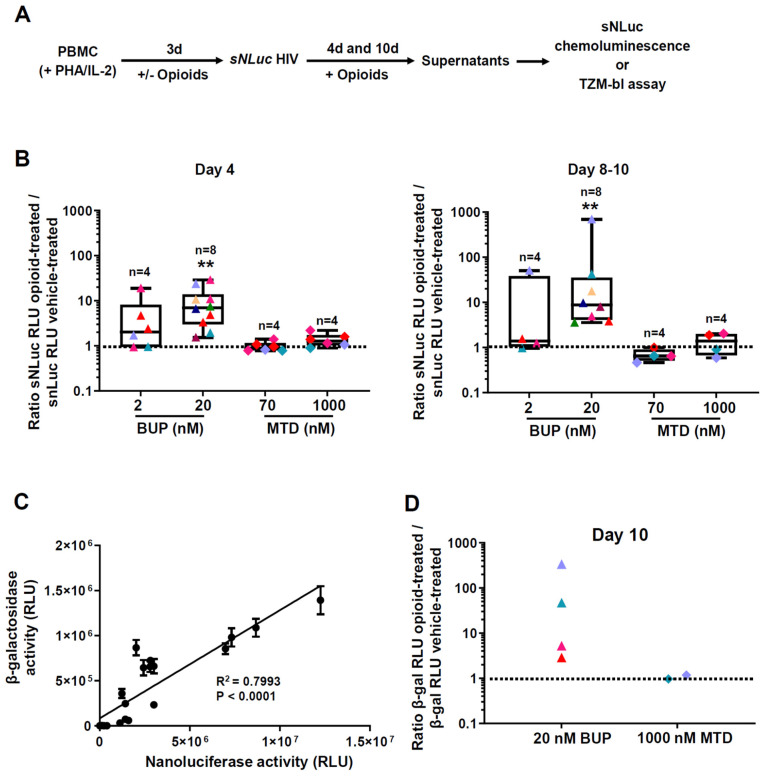
Comparison of methadone and buprenorphine on HIV-1 propagation in longer-term PBMC cultures. (**A**) Experimental approach to assess in vitro HIV-1 infection and replication. PBMC were incubated with opioids for 72 h, infected and washed, then the opioid drugs were added back to the culture media. Aliquots were taken at the indicated timepoints, and HIV-1 replication was measured by quantifying nanoluciferase activity in the supernatants, and by infecting Tzm-bl cells and measuring β-gal activity. Chemiluminescence signals were quantified using a luminometer. (**B**) Summary plots showing the effect of the two opioids on nanoluciferase activity measured in supernatants taken at days 4 and 8–10 after HIV-1 infection. The results shown are for a MOI of 0.5. Each donor is represented by a different color; buprenorphine (BUP) and methadone (MTD) treatments are represented by triangles and rhomboids, respectively. The number of different donors per treatment is indicated with an “*n*” above the plots. We did between 4 and 12 technical replicates per condition. *X*-axis indicates different treatments and concentrations. *Y*-axis shows the ratio of luciferase activity measured in opioid-treated cultures over luciferase measured in vehicle-treated cultures. Plots indicate median (vertical lines in the middle), interquartile range (boxes), and min to max range (whiskers). *p*-values were two-tailed and calculated using the Wilcoxon Signed Rank Test, comparing the median obtained in each treatment with a theoretical median value of 1. **: *p* < 0.01. (**C**) Correlation between nanoluciferase and beta-galactosidase (β-gal) enzymatic activities. PBMC from two different donors were infected with 0.5 MOI sNLuc HIV-1. Supernatants were collected on day 10 after infection. Nanoluciferase activity (*x*-axis) was determined in a 20 µL fraction of the sample (total = 23 determinations). In parallel, TZM-bl cells were infected with a 1:500 dilution of another aliquot of the same samples. Infections were done using five technical replicates. Each point represents the average of these replicates +/− 1 SD. β-gal activity (*y*-axis) was measured 48 h post TZM-bl infection. *p* < 0.0001 for the linear regression. RLU: Relative Light Units. (**D**) Comparison of β-gal activities when PBMC were treated with 20 nM buprenorphine versus 1 µM methadone. The data were generated from four different donors. β-gal activity was measured after infecting TZM-bl cells with supernatants (1:50 dilutions) collected 10 days post 0.5 MOI sNLuc HIV infection. Number of replicates (independent infections) done per supernatant ranged between 10 and 20. *X*-axis shows different treatments; *Y*-axis indicates the ratio of β-gal activities measured in opioid-treated samples versus vehicle-treated controls. *p* = 0.0667 by Mann–Whitney one-tailed test.

**Figure 5 viruses-13-01472-f005:**
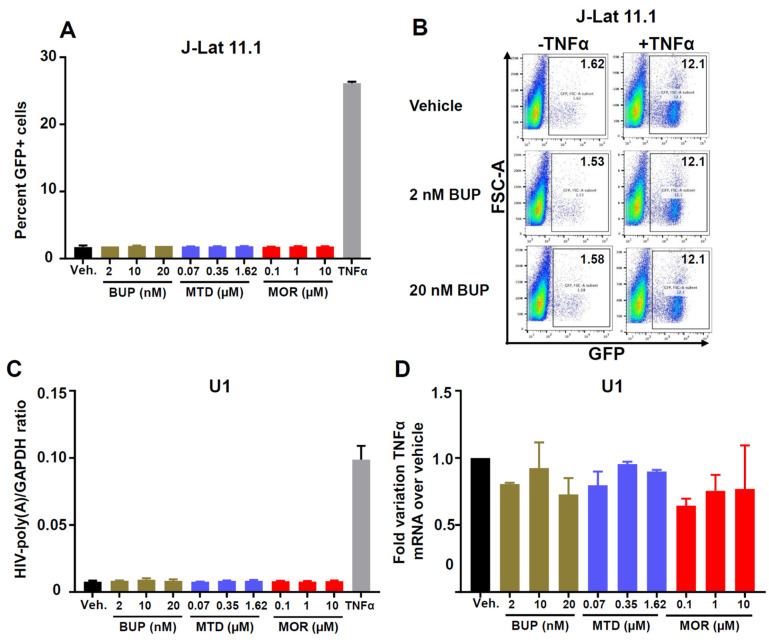
Effect of buprenorphine, methadone and morphine on HIV-1 reactivation in the HIV-1 latently infected T cell line J-Lat and the monocytic cell line U1. (**A**) J-Lat cells (clone 11.1) were treated for 72 h with vehicle, buprenorphine (2, 10, or 20 nM), methadone (70, 350, or 1620 nM) or morphine (0.1, 1, or 10 µM). TNF-α (10 ng/mL) treatment was included as a positive control for reactivation. GFP expression was evaluated in live (NearInfraRed^negative^) cells by flow cytometry. *Y*-axis shows % GFP^+^ cells. Bars indicate the average of two technical replicates + 1 SD. These results are from a representative experiment repeated three times. (**B**) J-Lat cells were treated for 4 days with either vehicle, 2 or 20 nM buprenorphine, and later activated for 24 h with 1 ng/mL of TNF-α. Numbers in the upper right corners denote % GFP^+^ cells. *Y*-axis, scatter signal; *X*-axis, GFP fluorescence. This experiment was done with two technical replicates and only the dot plots from one of the duplicates are shown. (**C**) HIV-1 reactivation in U1 cells. These monocytic cells were treated for 72 h with the indicated opioids (or TNF-α) and RNA was isolated and converted into cDNA for reactivation analysis. *Y*-axis: GAPDH-normalized HIV-LTR-poly(A) copies, as determined by RT-droplet digital (ddPCR) assay. *X*-axis: different treatments. Bars indicate the average of two technical replicates + 1 SD. These results are from a representative experiment repeated three times. (**D**) TNF-α mRNA copies in opioid-treated U1 cells. Treatment with the indicated drugs was done for 72 h. RNA was extracted and converted to cDNA. TNF-α levels were determined by real-time PCR (qPCR). Relative quantification was done using the ΔΔCt method and GAPDH as a normalizer gene. *Y*-axis shows the fold variation over the vehicle-treated control. Bars indicate the average of two technical replicates + 1 SD.

## Data Availability

Data is contained within the article or Appendix A. Additional information is available on request from the corresponding author.

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
