# Peer review of "Buprenorphine Increases HIV-1 Infection In Vitro but Does Not Reactivate HIV-1 from Latency"

_viruses, 2021, doi:10.3390/v13081472_

Round 1

Reviewer 1 Report

It is hard to understand the article. The introduction is not sufficient, and the results section is challenging to follow. Only the method section is well-written. Overall, I feel the paper should be accepted after a significant revision.

My suggestions are here:

  1. Please enlarge the introduction and put some more information. For example, it is not clear which opioids suppress which immune characteristics on line 57.
  2. Without getting much information, it is unclear what opioids were used to affect the HIV-1 life cycle positively.
  3. Please be consistent in the method section regarding the culture of different cells. You should provide temperature and CO2 concentration for all cells.
  4. It is unclear why different MOI is used in Fig. 3B (between lines 508 and 514).
  5. Some of the words in the figures are not at the proper resolution.
  6. Is there any cells that don’t express NOP receptor and you can include them in the experiments?
  7. A better version of the second type of experiment should be done by keeping the opioid in the culture throughout the experiment. I will suggest doing this experiment in place of replenishment of opioids after the initial drug-free culture.

Author Response

We thank the reviewers for their insightful comments, which helped us to improve the clarity and quality of our report. We are pleased to submit a revised version, in which we have responded to all of the reviewers’ queries. We hope the paper will now be acceptable for publication in Viruses.

Reviewer # 1

It is hard to understand the article. The introduction is not sufficient, and the results section is challenging to follow. Only the method section is well-written. Overall, I feel the paper should be accepted after a significant revision.

My suggestions are here:

Please enlarge the introduction and put some more information. For example, it is not clear which opioids suppress which immune characteristics on line 57.

Thank you for your suggestions. We revised this part and clarified this point in the updated manuscript.

Without getting much information, it is unclear what opioids were used to affect the HIV-1 life cycle positively.

In the Introduction, we mentioned that both heroin and morphine can enhance HIV replication (lines 48-49). In line 80, we mentioned that MAT drug methadone also has a positive effect on HIV-1 life cycle.

In the Discussion, we expanded the mechanism for morphine (lines 760-773) and mentioned methadone again (line 775).

We did not include the paragraph below in the Introduction (or in the Discussion) because we think that neither natural peptides nor experimental synthetic agonists have current public health relevance:

“In addition to heroin, morphine and methadone, endogenous peptides such as beta endorphins can enhance the synthesis of HIV and reactivation of a latent infection via induction of proinflammatory cytokines in perivascular microglia (PMID: 7560019). Treatment of T cells with the synthetic μ agonist DAMGO also led to an enhancement of HIV infection, through upregulation of CCR5 (PMID: 12726730).”

Please be consistent in the method section regarding the culture of different cells. You should provide temperature and CO2 concentration for all cells.

We apologize for this omission and added this information for all the cell lines that we used.

It is unclear why different MOI is used in Fig. 3B (between lines 508 and 514).

Figure 3B shows the nanoluc results from only one donor, in which we tested the effects of morphine and buprenorphine on HIV-1 early replication. Here we showed the results for different MOIs 0.1, 0.2 and 0.5. This represents only one example and the purpose of this figure is to illustrate that the enhancement in HIV-1 replication driven by buprenorphine was consistent at different ratios of viral particles/infected cells. We used the same range of MOIs (0.1 to 0.5) across different donors and different treatments. For simplicity, Figs. 3C and 4B show the results for only one MOI. 

Some of the words in the figures are not at the proper resolution

We agree with the reviewer and replaced the figures in the new Word file with new ones with higher resolution.  Please note that the PDF original file uploaded onto Viruses website [PDF named: usse_submitted_to_Viruses,_04.27.21_.pdf, uploaded on 04.27.2021] had the figures with high resolution. We think that during the editorial formatting (when the final Word file: “viruses-1219772” was created), there was a substantial decrease in the resolution of the figures.

Are there any cells that don’t express NOP receptor and you can include them in the experiments?

Our results show that all the leukocytes analyzed (bulk PBMC, blood fractions enriched in monocytes or lymphocytes, blood-purified CD4+ and CD8+ T cells as well as intraepithelial T cells and macrophages) express NOP receptor. We also found that NOP is expressed in epithelial cells, which are not susceptible to HIV infection. Our results match those deposited in The Human Protein Atlas: NOP is expressed in a wide range of HIV target cells such as total PBMCs, monocytes, T cells and dendritic cells. We think that rather than analyzing cells that do not inherently express NOP to assess the functional role of this receptor and its ligands, we would have to knock it down (siRNA), genetically ablate it (e.g., CRISPR) or pharmacologically block it (e.g., LY2940094) on HIV susceptible cell types like T cells or macrophages.  We believe these are great experiments for a follow-up paper that will shed some light into the function on NOP receptor in HIV/AIDS pathogenesis.

A better version of the second type of experiment should be done by keeping the opioid in the culture throughout the experiment. I will suggest doing this experiment in place of replenishment of opioids after the initial drug-free culture.

We have already done this proposed experiment and the results are shown on Figure 4. PBMCs were cultured for 3 days in the presence of PHA, IL-2 and opioids (or vehicle control), so drugs were included during this initial culture (for clarity, please refer to schema, shown on Fig. 4A). After counting the cells, we did the infection with the nanoluc HIV vector (for 2.5h) in media without opioids. We washed the cells and immediately resuspended them in media containing opioids. Therefore, in these “long-term” experiments, opioids were present all the time from the day of thawing the cells to the endpoint, with the short ~3h interruption during HIV infection and washes.

Reviewer 2 Report

In this paper, Gornalusse et al. aimed to address the impact of medication-assisted drugs on HIV infection and HIV reactivation. This is a very relevant topic given the opioid epidemic and its overlap with the HIV epidemic. However, some concerns need to be addressed.

1- this study is all based on in vitro experiments and cell lines. Why are basically all figures n=2? There are no statistics presented to endorse the statements in the paper. Increasing the number of donors will allow proper statistical analysis. The majority of the figures showed technical replicates, but the manuscript needs biological replicates to reach a more robust conclusion. 

2-Figure 1:
    -Dose of Buprenorphine used was 2nm. Later in the paper authors used 2nM and 20nM. Does 20nM impact the receptor expression?

3- Figure 2 could be moved to supplement. 

4- Figure 3:
    - MOR and BUP have different vehicle controls (methanol and water), so the graphs should represent both vehicle controls. Also, what is the cell viability after adding methanol to them? One good addition to the paper would be showing cell viability after treatment with all drugs used in the paper.
    -Legend says that Panel B is n=1 donor. Is the symbol in the graphs technical replicates? Once again, authors need biological replicates too, and not only technical replicates. 
    -Panel C is not in the paper.

- Figure 4: 
    - I think it would be better to show the results in panel B as raw data instead of the ratio. This is the paper's main result, and presenting the raw data would allow the reader to check background RLU from the vehicle. 
    - Correlation between technical replicates will not be informative since they are from the same donor, excluding the intrapatient variability. It should be done in a larger number of donors and then correlate results from 2 methods for different donors. 

- Figure 5: 
    - why the concentrations of TNFa from panels A and B are different? In order to compare the reactivation data, the drug concentration needs to be consistent between the experiments.
    - Different clones from JLat behave differently upon stimulation and treatment. Did you try other clones? Was the result similar?
    - Graphs on panel B seem to be the same. Maybe because of the low quality of the image, it is hard to see differences, but it would be good to look at that.

- Figure 6:
    - Panel A is not necessary. Showing delta ct without any other statistical analysis does not mean much. Also, in the experiment, the cells were not activated with PHA and IL2 as in previous experiments. To address the result observed in figure 4 and check if the CCR5 is related to the result, the CCR5 flow data should have been done after activating cells. Basically same design from figure 4A but measuring the CCR5 receptor in the end. 
    - Panels C, D, and E, the biological replicates should be presented in the same graph, as it was done in figure 4. This figure is a perfect example of why the biological replicates from the entire paper need to be increased: 20nM of BUP, 72hPHA+opioid, donor one proliferation is ~30%. Donor 2 is less than 10%.  The statement that drugs did not affect proliferation can not be based on only two donors, with considerable variability between them. 
    - It seems like 20nM BUP is better based on Figure 4. Why on panels F and G 2nM was used? Maybe the difference would appear in the higher dose.  Also, those are only n=2, the p-value from n=2 is not accurate. 

- For all figures, please enhance the quality. 

- Keep MOI consistent along with all experiments. 

Author Response

Reviewer # 2

1- this study is all based on in vitro experiments and cell lines. Why are basically all figures n=2? There are no statistics presented to endorse the statements in the paper. Increasing the number of donors will allow proper statistical analysis. The majority of the figures showed technical replicates, but the manuscript needs biological replicates to reach a more robust conclusion. 

We apologize for this omission. However, please note that the PowerPoint file containing the figures [emailed to Milan Milicevic on 04.28.2021, named “FINAL Figures Opioid Paper 04.27.2021 (corrected for PDF)] as well as the ZIP file uploaded in the Viruses website [PDF named: usse_submitted_to_Viruses,_04.27.21_.pdf, uploaded on 04.27.2021] did contain Fig.3C. We think that this dataset probably got lost due to editorial reformatting while the final Word file (“viruses-1219772”) was being created. We noticed even a light gray line above the legend for figure 3 (line 552), suggesting that it was inadvertently cut.

In Figure 1, we did not look at functional read-outs, but we checked which cells expressed the opioids receptors. We think that the experiments done are sufficient to illustrate the range of expression of NOP receptor. For example, NOP mRNA was analyzed in 4 different PBMC donors [2 unfractioned, (Fig.1A) and 2 fractioned, (Fig.1B)]. Likewise, NOP mRNA was found in 3 vaginal samples (2 shown in Fig. 1D and 1 in Fig. 1E).

It is plausible that different expression levels of NOP can influence the results for inter-donor variability in HIV infection. To figure that out, we should measure quantitatively the levels of NOP, in parallel to infections; this is a more involved study. Just adding a third donor will not be enough to explain these differences. We have just started conducting those experiments for a follow-up paper.

Figure 3C and 4B are from five or more separate individuals. The number of different donors (“n”) are now indicated in the figures. These are core data and we did repeat several biological replicates for those findings we found most important. The most significant findings related to enhancing effect elicited by 20 nM buprenorphine were confirmed in 10 and 8 independent experiments (for days 4 and 8-10, respectively) conducted on 8 different donors, as shown in Fig. 4B, left and right panels. To also assess the intra-donor variability (between days), for some donors we performed different cultures and HIV infections on separate days and represented the data with the same symbols and colors. For each condition (drug, concentration, HIV MOI) we included at least 3 technical replicates. The statistical tests conducted for the summary of the core data are described in the figure legends.

2-Figure 1:
    -Dose of Buprenorphine used was 2nm. Later in the paper authors used 2nM and 20nM. Does 20nM impact the receptor expression?

We measured NOP expression only after adding 2 nM BUP. However, the purpose of Fig. 1A was to show that PBMC express this NOP receptor and not that BUP affects the expression of opioids receptors. To make it clear, we remade Fig. 1A and left out the data with 2 nM BUP.

3- Figure 2 could be moved to supplement. 

We think that because this reporter construct is new and unpublished, it merits its inclusion in the main text and not in the supplement.

4- Figure 3:
    - MOR and BUP have different vehicle controls (methanol and water), so the graphs should represent both vehicle controls. Also, what is the cell viability after adding methanol to them? One good addition to the paper would be showing cell viability after treatment with all drugs used in the paper.

We thank the reviewer for this suggestion. In Fig. 3B (and Fig.3C), we compared the short-term effects of buprenorphine versus morphine. In these experiments, the vehicle control was consistently made with R10 media with PHA, IL-2 and 1.25% V/V of methanol. It contains the equivalent amount of methanol present in the 100 µM morphine media (1/80 dilution of the 8 mM stock). In each experiment, shown in Figure 3, we determined the concentration of live cells for each condition, either by trypan-blue staining or using Guava Muse® analyzer.  The viability in the vehicle containing 1.25% methanol had a mean of 61.12% (range: 51.3-80.0%, n=5) and for other treatments was similar and indicated in Supplemental Fig 1A (please see at the end of the revised manuscript for all Supplemental Figures). This range of variability is normal 3 days after PHA stimulation. Please note that the viability in vehicle control was not lower than those measured in 2nM or 20nM BUP. In the experiments shown in Figure 4, in which we compared the long-term effects of buprenorphine and methadone, when both drugs are present during the course of the incubation, the vehicle control was R10 media with PHA and IL-2. Again, the viability was very similar across conditions. We included a Supplemental Figure 1B to show this point.

    -Legend says that Panel B is n=1 donor. Is the symbol in the graphs technical replicates? Once again, authors need biological replicates too, and not only technical replicates. 
    -Panel C is not in the paper.

Fig. 3B shows the data for n=1 donor. Each symbol represents a technical replicate. For this particular experiment, we set up n=3 replicates for MOI 0.1 and 0.5, and n=6 replicates for MOI 0.2. The purpose of this panel was to illustrate the variability between technical replicates and to show that the enhancement effect of buprenorphine was consistent across MOIs.

As indicated in the point 1 (please see above, on page 2), the core data showing all the biological replicates derived from different donors are in Fig. 3C (short-term) and 4B (long-term).  We apologize for the omission of panel C but we believe this error happened during the editorial reformatting. Both the PDF (ZIP file) uploaded in Viruses portal and the PowerPoint and Word files emailed to the Assistant Editor (Mr. Milicevic, on April 28, 2021) contained the Fig. 3C. This figure shows the data for different donors, some of whom were repeated on different days (represented with same symbol and color). We did a minimum of 6 and a maximum of 13 independent experiments, with a minimum of 4 and a maximum of 6 independent donors. The number of technical replicates for the MOI 0.2 shown in Fig. 3B (shown as separate replicates) and 3C (shown as ratio of replicates) was 6.  We applied a statistical test (indicated in the figure legend) and indicated the statistical significance with asterisks (and the actual p-value in the text).

- Figure 4: 
    - I think it would be better to show the results in panel B as raw data instead of the ratio. This is the paper's main result, and presenting the raw data would allow the reader to check background RLU from the vehicle. 
    - Correlation between technical replicates will not be informative since they are from the same donor, excluding the intrapatient variability. It should be done in a larger number of donors and then correlate results from 2 methods for different donors. 

We agree with the reviewer and include the raw data as a new Supplemental Figure 2.  We kept panel B in the main text unchanged to be consistent with Fig. 3C. The number of independent experiments ranged between 4 to 10, and independent donors, between 4 to 8. The number of technical replicates ranged between 4 and 12. Donors tested on different days are represented with same colors. In 4B, again, we applied a statistical test (indicated in the figure legend) and indicated the statistical significance with asterisks (and the p-values in the main text).

The Fig. 4C shows the correlation of the read-outs measured by different techniques. The purpose of this panel was to show the concordance between nanoluciferase activity and the concentration of HIV particles, measured by TZM-bl infectivity assay. As indicated in the figure legend, this dataset was generated from 2 different donors, with 5 different β-galactosidase technical replicates. The aim of 4C was to show the accuracy of the determination and not the variability between biological replicates.

The most important message of the paper (long-term enhancement effect on HIV replication elicited by 20 nM BUP) was confirmed by including 8 different donors in Fig. 4B (nanoluc) and 4 donors in Fig. 4D (β-gal).

- Figure 5: 
    - why the concentrations of TNFa from panels A and B are different? In order to compare the reactivation data, the drug concentration needs to be consistent between the experiments.
    - Different clones from JLat behave differently upon stimulation and treatment. Did you try other clones? Was the result similar?
    - Graphs on panel B seem to be the same. Maybe because of the low quality of the image, it is hard to see differences, but it would be good to look at that.

In Fig. 5A, we used 10 ng/ml TNF-α because we used it as a standalone positive control for reactivation. In our previous paper (DOIhttps://doi.org/10.1128/JVI.01904-19, see Fig 1B and 1C in that JVI paper), we observed that this is the minimal concentration of TNF-α that induces the maximal reactivation in the J-Lat 11.1 line. In contrast, in Fig. 5B, we used 1 ng/ml (which is suboptimal for reactivation) because we wanted to answer the question whether buprenorphine has additive or synergistic effects when there is a lower level of this inflammatory cytokine.

We chose to analyze potential reactivation effects on J-Lat 11.1 because: (1) In our previous work, we showed it has less spontaneous reactivation than other J-Lat cells (~10% GFP+ in media), but it is more easily reactivatable by latency-reversing agents, including TNF-α (please, again, see Fig. 1C from DOIhttps://doi.org/10.1128/JVI.01904-19). Based on our previous testing, this cell line is even more sensitive to reactivation than CD4+ primary cells. (2) J-Lat 11.1 possess an HIV structure more similar to the HIV provirus found in vivo, compared to some of the other J-Lats like A1 and A7, which contain only a mini-HIV cassette.

We recognize that it is feasible that J-lats different from 11.1 react different to opioids, because they have HIV inserted in different genomic loci.  We have done a preliminary experiment (added as Supplemental Figure 3), in which we treated J-Lat A7 with increasing doses of buprenorphine and we have not seen reactivation with this drug in that cell line either, even at a 2 µM dose. This experiment was done with 2 technical replicates for each dose, and included TNF-α as a positive control. We are happy to do similar reactivation experiments in additional J-Lats if the reviewer considers it necessary.  It will take us more than 10 days to obtain these data but we would be willing to include it in a revised version of the manuscript.

We apologize for the low resolution. The data were obtained using 2 technical replicates per condition and coincidentally, the percentages were 12.1% GFP+ cells for all three conditions.  We hope with the revised figure, the reviewers can distinguish the individual points and see that they are different between the three conditions. .

- Figure 6:
    - Panel A is not necessary. Showing delta ct without any other statistical analysis does not mean much. Also, in the experiment, the cells were not activated with PHA and IL2 as in previous experiments. To address the result observed in figure 4 and check if the CCR5 is related to the result, the CCR5 flow data should have been done after activating cells. Basically same design from figure 4A but measuring the CCR5 receptor in the end. 
    - Panels C, D, and E, the biological replicates should be presented in the same graph, as it was done in figure 4. This figure is a perfect example of why the biological replicates from the entire paper need to be increased: 20nM of BUP, 72hPHA+opioid, donor one proliferation is ~30%. Donor 2 is less than 10%.  The statement that drugs did not affect proliferation can not be based on only two donors, with considerable variability between them. 

We agree with the reviewer. Fig. 6 shows premature data and is not satisfying overall because it was only obtained with n=2 different donors. We decided not to include Fig. 6 in the revised manuscript. We are trying to prove negative findings, i.e. mechanisms not involved in buprenorphine’s effect on HIV replication. To conclusively prove negative data, we would have to do at least 8-10 different donors.

    - It seems like 20nM BUP is better based on Figure 4. Why on panels F and G 2nM was used? Maybe the difference would appear in the higher dose.  Also, those are only n=2, the p-value from n=2 is not accurate. 

We decided to withdraw Fig. 6, as explained in the previous point.

- For all figures, please enhance the quality. 

The revised manuscript has figures with better quality.

- Keep MOI consistent along with all experiments. 

The most relevant findings pertaining to buprenorphine effect were obtained with similar MOIs. Fig. 3C shows 0.2 and 4B, 0.5.

Round 2

Reviewer 1 Report

Kindly accept the article. They made the changes and replied all questions.  Thanks for sharing the word file. 

Reviewer 2 Report

The edits made to the paper, and the addition of the supplements figures enhanced the manuscript positively. One last comment, the information about the rationale why choosing JLat 11.1 was very useful to me and can also be useful to the reader. Adding that information with reference (reference about 11.1 being similar to HIV provirus in vivo) would be a great addition. 

Author Response

Thank you very much for your comment. We followed your advice and added sentences to lines 534 to 540.